# Adapter CAR T Cell Therapy for the Treatment of B-Lineage Lymphomas

**DOI:** 10.3390/biomedicines10102420

**Published:** 2022-09-28

**Authors:** Daniel Atar, Anna-Sophia Mast, Sophia Scheuermann, Lara Ruoff, Christian Martin Seitz, Patrick Schlegel

**Affiliations:** 1Department of Pediatric Hematology and Oncology, University of Tuebingen, 72076 Tuebingen, Germany; 2DFG Cluster of Excellence 2180 Image-Guided and Functional Instructed Tumor Therapy (iFIT), University of Tuebingen, 72076 Tuebingen, Germany; 3School of Medical Sciences, Faculty of Medicine and Health, University of Sydney, Sydney 2006, Australia; 4Cellular Cancer Therapeutics Unit, Children’s Medical Research Institute, Sydney 2145, Australia; 5Department of Pediatric Hematology and Oncology, Westmead Children’s Hospital, Sydney 2145, Australia

**Keywords:** adapter CAR T cell, immune escape, combinatorial immunotargeting, B-cell lymphoma

## Abstract

CD19CAR T cells facilitate a transformational treatment in various relapsed and refractory aggressive B-lineage cancers. In general, encouraging response rates have been observed in B-lineage-derived non-Hodgkin’s lymphomas treated with CD19CAR T cells. The major cause of death in heavily pretreated NHL patients is lymphoma progression and lymphoma recurrence. Inefficient CAR T cell therapy is the result of the limited potency of the CAR T cell product or is due to loss of the targeted antigen. Target antigen loss has been identified as the key factor that can be addressed stringently by dual- or multitargeted CAR T cell approaches. We have developed a versatile adapter CAR T cell technology (AdCAR) that allows multitargeting. Screening of three different B-lineage lymphoma cell lines has revealed distinct immune target profiles. Cancer-specific adapter molecule combinations may be utilized to prevent antigen immune escape. In general, CD19CAR T cells become non-functional in CD19 negative lymphoma subsets; however, AdCAR T cells can be redirected to alternative target antigens beyond CD19, such as CD20, CD22, CD79B, and ROR-1. The capability to flexibly shift CAR specificity by exchanging the adapter molecule’s specificity broadens the application and significantly increases the anti-leukemic and anti-lymphoma activity. The clinical evaluation of AdCAR T cells in lymphoma as a new concept of CAR T cell immunotherapy may overcome treatment failure due to antigen immune escape in monotargeted conventional CAR T cell therapies.

## 1. Introduction

Therapies based on genetically engineered T cells to express a chimeric antigen receptor (CAR) have revolutionized the treatment of relapsed and refractory patients with B-lineage cancers [1]. For the treatment of B-lineage-derived cancers, a total of four CD19- and two BCMA-targeted CAR T cell products with distinct biological properties have been approved by the US FDA over the last 5 years since 2017 and have contributed to improved survival in acute lymphoblastic leukemia, non-Hodgkin’s lymphoma, and multiple myeloma [2].

The major acute complications of CD19 CAR T cell therapies are cytokine release syndrome (CRS), immune effector cell-associated neurotoxicity syndrome (ICANS), and macrophage activation syndrome (MAS) originating from T cell activation and proliferation with excessive secretion of cytokines and triggering of an immune activation cascade [3]. Despite growing experience in the management of these life-threatening acute toxicities, patients still frequently require intensive care treatment [4]. However, the major causes of death in CAR T cell-treated patients remain leukemia, lymphoma, and multiple myeloma recurrence due to treatment failure [5]. The main drivers of CAR T cell therapy failure are directly or indirectly linked to the targeted antigen. Target antigen downregulation, point mutations, alternative splicing, and biallelic loss of the antigen have been identified as the drivers of resistance to CD19- and BCMA-CAR T cell therapy in up to 40% of patients [6,7,8,9,10]. Indirect failures are caused by CAR T cell dysfunction while the targeted antigen remains present. In multiple myeloma, antigen shedding and trogocytosis have been shown to induce CAR T cell fratricide. Moreover, the upregulation of immune checkpoint ligands such as (PD-L1) and the hypoxic niche in the bone marrow limit CAR T cell function [11].

We have developed an adapter CAR technology (AdCAR) that allows functional control of the CAR-expressing cells via the introduction of adapter molecules [12,13]. AdCAR T cells are functionally inert without the adapter molecules and mediate anticancer function only in the presence of adapter molecules [13]. Direct effects are defined as downregulation or loss of the targeted epitope of an antigen. Indirect failures are mechanisms in which the CAR T cell function is impaired while the targeted antigen remains present, such as antigen shedding, overstimulation by fratricide, or targeting of healthy tissues. The vast majority of target antigen-dependent CAR T cell treatment failures can be addressed highly efficaciously by multitargeted approaches.

We hypothesized that our AdCAR T cells could target B-lineage lymphomas via adapter molecules targeted to CD19 and alternative antigens present on non-Hodgkin’s lymphoma. The advantage over conventional direct CD19 CAR T cells is the possibility to broaden the targets via alternative antigens beyond CD19 and perform combinatorial as well as sequential targeting. We compared the in vitro function of CAR T cells by different measures with a second-generation CD19 CAR and a third-generation AdCAR construct with regards to specific CAR T cell activation, cytokine secretion, cytolysis, and exhaustion. The key feature of AdCAR T cells compared with conventional CAR T cells is the ability for multitargeting, which may overcome antigen immune escape, which is responsible for treatment failure with conventional CD19 CAR T cells. We provide a proof of concept for our AdCAR technology as a platform technology for adoptive T cell therapy to address the major cause of treatment failure with CAR T cells in B-lineage lymphomas.

## 2. Material and Methods

### 2.1. Design of CAR Constructs and Lentiviral Vectors

A CD19 single chain variable fragment (scFv) was derived from the murine anti-CD19 mAb clone 4G7. The AdCAR anti-LLE-biotin scFv was derived from the murine mAb clone mBio3. The scFvs were designed in light–heavy (LH) configuration with a standard G4S3 linker. The CD19 was used in a second-generation CAR backbone comprising a IgG4 hinge, a CD8 transmembrane domain, the cytoplasmic costimulatory domain of 4-1BB, and the CD3ζ signaling domain. Both constructs co-expressed truncated epidermal growth factor receptor (tEGFR) downstream from a ribosomal skip P2A site. The custom gene synthesis of the CAR constructs was carried out by GeneArt (ThermoFisher, Waltham, MA, USA). The CAR constructs were subcloned into a second-generation lentiviral transfer vector for generating self-inactivating lentiviral particles (SIN-LV). SIN-LV were produced in Lenti-X 293T cells (Takara) after lipofection (Lipofectamin 3000, ThermoFisher, Waltham, MA, USA) according to the manufacturer’s instructions utilizing a second-generation packaging plasmid, the VSV-G envelope plasmid, and the corresponding transfer plasmid for expression of the CD19 or the AdCAR (LLE-CAR). Supernatants containing SIN LV were harvested 24 h after lipofection, concentrated by the Lenti-X concentrator (TaKaRa, Kusatsu, Shiga, Japan) and cryopreserved according to the manufacturer’s instructions.

### 2.2. Isolation of Human Primary T Cells and Transduction

Peripheral blood mononuclear cells (PBMCs) were isolated from whole blood samples acquired from healthy volunteer donors at the University Children’s Hospital Tuebingen, by Ficoll centrifugation (Biocoll, Biochrom, Berlin, Germany). T cells were isolated by magnetic separation using anti-CD4 and anti-CD8 microbeads (Miltenyi Biotec, Bergisch-Gladbach, Germany) simultaneously. Isolated T cells were activated with TransAct^TM^ (Miltenyi Biotec) and cultivated in TexMACS media supplemented with 10 ng/mL IL7 and 5 ng/mL IL15 (Miltenyi Biotec, Bergisch-Gladbach, Germany). TransAct^TM^-activated T cells were transduced at a multiplicity of infection (MOI) of 3 after 36 h. Transduced T cells were maintained at 0.5–2 × 10^6^ cells/mL in IL7/IL15-supplemented TexMACS^®^ media. On Days 7+, CAR transduction efficiency and the CD4/CD8 ratio were determined by flow cytometry. 

### 2.3. Cell Line Culturing

All lymphoma cell lines (JeKo-1, Raji, Daudi) were purchased from ATCC (Manassas, VA, USA) or DSMZ (Braunschweig, Germany), and Lenti-X 293T cells were purchased from (TaKaRa, Kusatsu, Shiga, Japan). All lymphoma cell lines were maintained in RPMI 1640 media and Lenti-X 293T cells were maintained in DMEM media. Media were supplemented with 10% heat-inactivated fetal bovine serum (ThermoFisher, Waltham, MA, USA), 2 mM L-glutamine, 1 mM sodium pyruvate, and 100 units/mL of penicillin and 100 µg/mL of streptomycin (all from Biochrom, Waltham, MA, USA), referred to as complete media.

### 2.4. Generation of Luciferase-Expressing Cell Lines

Transfer plasmids, based on a third-generation lentiviral transfer vector plasmid encoding firefly luciferase and green fluorescent protein (GFP), were kindly provided by Irmela Jeremias, Helmholtz Center Munich, Germany. LV particles were generated as described above. Cell lines were transduced at a MOI of 3. Transgene expression was confirmed by flow cytometry. Transduced cells were enriched by bulk fluorescence-activated cell sorting.

### 2.5. Generation of Knockout Cell Lines Using CRISPR/Cas9 Technology

All gRNAs were designed with the online tool “CHOPCHOP” (http://chopchop.cbu.uib.no/, accessed on 16 April 2020) and synthesized by ThermoFisher Scientific. To form a CRISPR/RNP complex that was ready to transfect, the gRNA and the trans-activating CRISPR RNA (tracrRNA) (ThermoFisher, Waltham, MA, USA) were annealed directly before use. The tracrRNA and all gRNAs were dissolved in an appropriate amount of TE buffer (10 mM Tris, 0.1 mM EDTA; pH 7.5) to get working solutions of 100 pmol/µL. These working stocks were aliquoted to avoid freeze–thawing cycles and stored at −20 °C until use. The gRNA and the tracrRNA were mixed in an equimolar concentration in 1× annealing buffer (5× annealing buffer: 30 mM HEPES, 100 mM potassium acetate, and 2 mM magnesium acetate). For each transfection of 2 × 10^6^ cells, 75 picomoles of gRNA and 75 picomoles of tracrRNA were used. The annealing reaction was carried out with a temperature gradient in a thermal cycler (95–25 °C; −0.1°/s). The annealed product (gRNA: tracrRNA) was mixed in an equimolar concentration with TrueCut Cas9 Protein v2 (ThermoFisher, Waltham, MA, USA) and incubated for at least 10 min and for a maximum of 30 min at RT to form the CRISPR/RNP complex. Transfection of the RNP was achieved with the Neon™ Transfection System 100 µL Kit according to the manufacturer’s instructions. Knockout cells were enriched by bulk fluorescence-activated cell sorting.

### 2.6. Luciferase-Based Cytotoxicity Assay (LCA)

Tumor cells were plated in complete RPMI media in white 96-well flat-bottomed plates (Greiner Bio One, Kremsmünster, Austria) with 30,000 cells per well. Synthetic D-luciferin (Sigma Aldrich, St. Louis, MO, USA) was added to each well at 4 µg/mL. Effector cells were plated at the indicated effector to target ratio (E:T). The total volume per well was 200 µL. Therapeutic antibodies or combinations thereof were used at the indicated concentrations. Plates were incubated in a HERAcell incubator (Heraeus, Hanau, Germany) at 37 °C, 95% humidity, and 5% CO_2_. Bioluminescence was measured using the Tecan SPARK microplate reader (Perkin Elmer, Waltham, MA, USA) at 37 °C, at the indicated time intervals. Lysis was calculated by the relative luminescence of the testing conditions according to a lysis formula based on a standard dilution series.

### 2.7. Immunophenotyping of Tumor Cell Lines 

Immunophenotyping was performed on a BD^TM^ LSR II flow cytometer. Antibody staining was carried out according to standard operating procedures at 4 °C in a PBS buffer. All commercial antibodies were purchased from (Miltenyi Biotec, Bergisch-Gladbach, Germany). The antibody clone is defined by the clone in brackets. The tumor-associated antigens were CD19 PE (REA675), CD20 PE (REA780), CD22 PE (REA340), ROR-1 PE (REA1051), CD276 PE (REA1094), CD79B PE (REA120), and CD10 PE (97C5). The activating ligands on tumor cells were CD112 APC (REA1195) and CD155 PE (REA1081) (DNAM-1 ligands), and MIC A/B APC (REA10876) (an NKG2D ligand). Indirect staining of therapeutic antibodies was achieved with Anti-Biotin PE (REA746). The isotype controls were IgG1 Isotype PE (REA293) and IgG1 Isotype APC (REA293). Antigen positivity was defined by staining of the tumor cells using primary labelled mAbs compared with the isotype control. Overton positivity was calculated by integral subtraction (specific fluorescence minus fluorescence of the isotype control) using FlowJo 10.4 software.

### 2.8. Quantification of Cytokine Release

Here, 2.5 × 10^5^ CAR T cells were cultivated with 2.5 × 10^5^ JeKo-1 cells at an ET ratio of 1:1 per well according to the indicated conditions in RPMI 1640-based complete media in a HERAcell incubator (Heraeus) at 37 °C, 95% humidity, and 5% CO_2_. Supernatants were collected after 24 and 120 h. Quantification of the cytokines was performed with cytokine capture beads using the MACSPlex^®^ custom cytokine assay with the indicated specificities. Data acquisition was carried out on a MacsQuant^®^ Analyzer 10 SN 2535 flow cytometer and with MACSQuantify^®^ software (2.11.1731.18902) according to the manufacturer’s instructions (Miltenyi Biotec, Bergisch-Gladbach, Germany).

### 2.9. Adapter Molecule Conjugation

Adapter molecule modification was performed at 21 °C for 1 h in 0.1 M NaHCO_3_ buffer using a 3-fold molar excess of biotin-LC-LC-NHS (ThermoFisher, CAS-No. 89889-52-1, Waltham, MA, USA), followed by separation of the antibody/label mix on a Sephadex G25 column. Protein-containing fractions were pooled, and the concentration was measured as the absorption at 280 nm. Successful conjugation was confirmed by LC-MS and/or by flow cytometry on cell lines expressing the target antigen and secondary staining with a fluorophore-conjugated anti-biotin antibody.

### 2.10. T Cell Immunophenotyping

The T cells’ maturation state was assessed via the expression of CD45RA, CD45RO, CD62L, and CD95. Five different states of maturation were distinguished: T_N_^i^, T_SCM_, T_CM_, T_EM_, and T_EMRA_. T naïve^i^ (T_N_^i^) cells were defined as (CD45RA^+^/CD45RO^−^/CD62L^+^/CD95^−^). We acknowledge that CD3/CD28 co-activated T cells per se are defined as non-naïve T cells. Thus, we classified T cells with a naïve immunophenotype as T_N_^i^. Stem cell-like memory T cells (T_SCM_) were defined as (CD45RA^+^/CD45RO^−^/CD62L^+^/CD95^+^), central memory T cells (T_CM_) were defined as (CD45RA^−^/CD45RO^+^/CD62L^+^CD95^+^), effector memory T cells (T_EM_) were defined as (CD45RA^−^/CD45RO^+^/CD62L^−^/CD95^+^), and effector memory T cells that re-expressed CD45RA (TEMRA) were defined as (CD45RA^+^/CD45RO^−^/CD62L^−^/CD95^+^). T cells’ activation state was assessed using CD25 and CD69. T cells’ exhaustion state was assessed by PD-1 expression. All commercial antibodies were purchased from Miltenyi Biotec. The antibody clone is defined by the clone in brackets. The T cell antigens were CD4 VioBlue (M-T466), CD8 APC-Vio770 (REA734), CD45RA VioGreen (REA1047), CD45RO VioGreen (REA611), CD45RO APC-Vio770 (REA611), CD62L PE (REA615), and CD95 APC-Vio770 (REA738). The CAR identification marker-truncated EGFR was measured via EGFR APC (REA688). The activating co-receptors on T cells were CD226 PE (REA1040) (DNAM-1) and CD314 APC (REA797) (NKG2D). The T cell activation markers were CD25 VioBright B515 (REA945) and CD69 VioBlue (REA824). The T cell exhaustion marker was CD279 PE (REA1165) [PD-1]. The isotype controls were IgG1 Isotype PE (REA293) and IgG1 Isotype APC (REA293).

## 3. Results

We designed our adapter CAR T cells (AdCAR T), which were targeted to biotin in the context of a specific linker structure, referred to as linker–label–epitope (LLE biotin) in a third-generation CAR T cell format. On the basis of our findings in previous studies, we learned that third-generation signaling was advantageous compared with second-generation signaling for AdCAR T cells [13]. The recognition domain was based on the murine anti-LLE-biotin clone mBio3.

The AdCAR-scFv was used in a light–heavy variable chain (LH) configuration. The CAR backbone consisted of an IgG4 hinge extracellular spacer domain variant, the CD8A transmembrane domain, CD28, and the 4-1BB costimulatory domains, as well as the CD3ζ signaling domain. Our CD19CAR was based on the commonly used murine FMC63 clone in LH-scFv configuration in a second-generation CAR backbone. The CD19CAR comprised an IgG4 hinge extracellular domain variant, the CD8A transmembrane domain, the 4-1BB costimulatory domain, and the CD3ζ signaling domain. A truncated version of the human epidermal growth factor receptor (tEGFR) was integrated downstream from a P2A site suitable for detection and enrichment of the CAR T cells. The CAR design and a schematic illustration of the CD19CAR, the AdCAR, and the tEGFR anchored to the cell membrane are shown in Figure 1a,b. The marker gene tEGFR allowed us to clearly identify the CAR T cell fraction of the whole T cell population by flow cytometry, as depicted in Figure 1c. The transgene expression of the CD19CAR T cell and AdCAR T cell constructs were comparable, indicating similar expression efficiencies.

### 3.1. Heterogeneity in the Antigen Expression Profiles of B-Lineage Non-Hodgkin Lymphoma

The essential features of adapter CAR systems include the possibility for transient, universal, simultaneous, and sequential multitargeting. To demonstrate the heterogeneity of B-lineage derived non-Hodgkin’s lymphoma, we immunoprofiled three cell lines: JeKo-1 (MCL), Raji, and Daudi (both Burkitt). Immunophenotyping was conducted using primary R-phycoerythrin (PE)-labeled mAbs (Figure 2) of the same fluorophore to estimate differential expression in a semiquantitative manner. The target antigen panel was chosen on the basis of the expression level found in B-lineage lymphomas and their suitability for CAR T cell therapy from our own experience and the literature. We focused on highly expressed antigens found in primary leukemia and lymphomas, such as CD19, CD20, CD22, ROR-1, CD276, CD79B, and CD10. Figure 2a shows the expression of the surface antigens in normalized histograms. The intratumoral heterogeneity is highlighted and visualized in two tables. One table shows a heat map with the fraction of antigen-positive cells as a percentage (Figure 2b), and the second table depicts the suitability of the target antigens for CAR-mediated targeting as a heat map (Figure 2c) by showing the median fluorescence intensity ratio (=MFIR). The intratumoral heterogeneity of antigens in CAR T cell therapy is best addressed by versatile multitargeted CAR technologies.

### 3.2. CAR-Specific Activation and Highly Specific Target Cell Lysis

In order to show the activation of AdCAR T cells strictly requires the presence of the AdCAR T cell, a specific adapter molecule, and a corresponding antigen-positive target cell, we performed co-incubation experiments and measured the upregulation of the early activation marker CD69 and CD25. The specifically activated CAR T cell subset was defined as double CD25^+^CD69^+^ positive. To test the antigen-specific activation of CAR T cells, we tested various combinations of effector cells, adapter molecules, and target cells. Only the combination of CD19CAR T cells in conjunction with the antigen-positive target cell line JeKo-1 activated CD19CAR T cells, and for AdCAR T cells, only the combination of AdCAR T cells plus the antigen-positive target cell line JeKo-1 plus an LLE-CD19 mAb adapter molecule activated the AdCAR T cells. The specific activation (CD25^+^CD69^+^ expression) of CAR T cells measured after 1 and 5 days was significantly increased compared with the unstimulated control. Neither under CD19CAR T cell conditions nor AdCAR T cell conditions was a relevant difference in the expression levels of CD25^+^CD69^+^ found after 1 and 5 days. Non-specific upregulation of CD25^+^CD69^+^ in AdCAR T cells in the presence of JeKo-1 was observed, whereas there was no significant non-specific upregulation of CD25^+^CD69^+^ detected in the presence of LLE-CD19 mAb at 1 ng/mL without the target cells (Figure 3a).

Cytokine secretion was assessed as a secondary indicator of CAR effector function. The cytokine concentrations were measured in the supernatants after 1 day and after 5 days (Figure 3b). The cytokine panel comprised the 12-cytokine multiplex panel: GMCSF, INFα, IFNγ, IL2, IL4, IL5, IL6, IL9, IL10, IL12, IL17a, and TNFα. With regards to the secretion kinetics of the most relevant CAR T cell cytokines GMCSF, IL2, IFNγ, and TNFα, we observed significantly higher cytokine levels secreted by conventional CD19CAR T cells than by AdCAR T cells after 1 day. However, after 5 days, significant differences in the cytokine profile of CD19CAR T cells and AdCAR T cells were no longer detectable. The antigen-specific CAR-mediated cytolysis is illustrated in Figure 4. Antigen-specific CAR activation was further supported by the activation-dependent cytokine secretion, a change in the immunophenotype to an effector cell phenotype T_EMRA_, and the activation-dependent expression of the exhaustion marker PD-1 (Figure 5).

To demonstrate the universal antigen-specific effector function of AdCAR T cells in vitro, we used adapter molecules in the LLE-mAb format targeted to various tumor-associated antigens expressed by the lymphoma cell lines Raji, Daudi (both Burkitt), and JeKo-1 (MCL). AdCAR T cells were incubated at six different effector-to-target (ET) ratios. The range covered 5:1 to 0.15:1, which corresponded to 1 CAR T cell versus 6.66 cancer cells. The LLE-mAb concentration was used at 1 ng/mL in all experiments per LLE-mAb. Therefore, combinations of LLE-mAb were used at the higher final LLE-mAb concentrations. The target cell lysis was determined by LCA, a luciferase-based cytotoxicity assay. In order to demonstrate the specificity of the AdCAR system, untransduced activated T cells served as negative controls. The LLE-CD19 mAb did not induce any measurable cytotoxic effects at 1 ng/mL. Conventional second-generation CD19CAR T cells served as positive controls and for benchmarking. Both monotargeting with LLE-CD19 only and combinatorial targeting with LLE-CD19 and LLE-CD20 were performed utilizing the wild-type form of the respective cell lines. The cytotoxic effect of the AdCAR T cells was significantly higher than the cytotoxic effect of activated T cells across all three NHL cell lines. However, combinatorial targeting using LLE-CD19 and LLE-CD20 did not outperform monotargeting in the wild-type cell lines (Figure 4a). To demonstrate the superiority of combinatorial targeting, we compared CD19/CD20 dual targeting with AdCAR T cells versus CD19 monotargeting with AdCAR T cells and CD19CAR T cells. Combinatorial CD19/CD20 targeting induced significantly higher cytolysis in the CD19 knockout variant of JeKo-1 (JeKo-1_CD19KO_) (Figure 4b). Moreover, to minimize non-specific lysis effects in JeKo-1 and in Raji, we demonstrated the capability of targeting alternative target antigens beyond CD19 at a low ET ratio of 1.25:1 (Figure 4c,d). It is noteworthy that some target antigens are more suitable for CAR T cell therapy than others. Targeting of different antigens elucidated different potencies, which were dependent on the antigen expression levels, but also the biological properties of the target antigens’ impact on the efficacy to recruit AdCAR T cells to the cancer cells. Thus, despite the capability for universal targeting (one CAR construct for all antigens), not all target antigens managed to recruit AdCAR T cells to cancer cells at the same level and mediate differential potencies. Additionally, we showed that combinatorial targeting of alternative antigens, such as CD10 in combination with ROR-1, or CD10 in combination with CD20, is feasible and can increase the targeting compared with monotargeting. In general, combinatorial targeting was superior compared with monotargeting in JeKo-1 and Raji (Figure 4c,d).

### 3.3. CAR T Cell Immunophenotype, Exhaustion, and CAR-Unrelated Anticancer Effects

At first, CAR T cell activation triggered the primary effector function cytolysis. Secondary effector functions included cytokine secretion support proliferation. Proliferation of CAR T cells induced a significant redistribution of immunophenotypic cell subsets. The composition of and change in the CAR T cell immunophenotype subsets served as indicators of tertiary CAR T cell functions, such as CAR T cell fitness and long-term persistence in patients.

The CAR T cell immunophenotype was assessed on Day 1 and Day 5 after co-incubation with the target cells to understand the physiological shift during the anticancer immune response. A clear shift of the CAR T cell subsets into mature effector T cells (T_EMRA_) was specifically induced by contact with target cells. The documented change in the immunophenotype composition to >70% T_EMRA_ plus around 20% T_EM_ cells was very similar for the conventional second-generation CD19CAR T cells and the third-generation AdCAR T cells after 1 day (Figure 5a). The registered T cell subsets’ composition after 5 days ws the result of lineage-maturation of cells and the subsequent disproportional proliferation of the CAR T cells maintaining a high proportion of T cell effector populations at 90% and above (Figure 5b). The changes in the immunophenotype indicate physiologic T cell maturation under CAR T cell engagement. In both CD19CAR T cells and AdCAR T cells, the proportion of reconstituting T_SCM_ was significantly reduced. Whereas the immediate maturation into effector cell subsets was required to control the cancer cells, the proliferation and maintenance of T_SCM_ facilitated reconstitution and persistence. 

Further, the exhaustion marker PD-1 was assessed by flow cytometry after 24 h (Figure 6a,b). AdCAR T cells were found to be exclusively activated in the presence of (1) antigen specific LLE-conjugated adapter molecules and (2) the corresponding antigen-expressing target cells. There was neither a difference in PD-1 expression between the CD19CAR T cells and the AdCAR T cells in the control groups nor under the testing conditions, in which PD-1 was upregulated from a baseline of around 5% to 30% PD-1 expressing cells after co-incubation with the target cells in the CAR T cell fraction of the cells.

The non-specific activation of the AdCAR T cells and cytolysis of the target cells without the LLE-CD19 adapter molecule is frequently observed in activated T cells and CAR T cells after simultaneous stimulation via the epsilon chain of the CD3 receptor complex and the costimulatory receptor CD28, which induces the expression of activating receptors, such as DNAM-1 and NKG2D [14]. The expression of DNAM-1 and NKG2D is indirectly supported by common γ-chain cytokines (IL2, IL7, IL15) on activated T cells and CAR T cells (Figure 7a) [15]. 

The abovementioned non-specific lysis of cancer cells by T cells and CAR T cells, which correspond to non-CAR-related anticancer effects in a NK-like mode of action, can be mediated by the activating receptors DNAM-1 and NKG2D. Due to the high expression of the NKG2D ligand MIC A/B in JeKo-1 and the high expression of CD314 in activated T cells (Figure 7b), the NK-like CAR independent recognition of JeKo-1 may be explained. To illustrate the effect of combinatorial targeting effect mediated by the CAR receptor with the AdCAR system, the E:T titration was repeated with a CD19 knockout variant of JeKo-1 (Figure 4b). At the low E:T ratios, the NK-like AdCAR activity is significantly reduced and the specific CAR-mediated effect becomes prominent.

To investigate the reason for the high non-specific activation and cytolysis of the AdCAR T cells targeting JeKo-1, we determined the surface expression of the activating coreceptors DNAM-1 and NKG2D on activated T cells. The cognate ligand expression for DNAM-1, CD112, and CD155 (DNAM-1L), as well as for NKG2D, the MHC Class I chain-related molecules A and B (MIC A/B; NKG2DL) were measured in JeKo-1 (Figure 7a,b). It appears that the high non-specific lysis of JeKo-1 and the long-term non-specific activation of AdCARs by JeKo-1 was due to the high expression of the NKG2D ligands MIC A/B on JeKo-1. Moreover, the degree of LLE labeling (DOL) per adapter molecule seemed to impact the non-specific activation of AdCAR T cells (Figure 7c). We co-incubated the AdCAR T cells without LLE-mAb (red) and with 10 ng/mL of various LLE-mAbs, which were either multibiotinylated (light grey) or monobiotinylated (dark grey), for 1 day (left) and for 5 days (right). Activation of CAR T cells after incubation was determined by flow cytometry by staining for the cell surface markers CD25 and CD69. The increase in the CD25^+^CD69^+^ double-positive AdCAR T cells was considered to be activated by the LLE-mAb. Overall, the upregulation of CD25 and CD69 was increased by LLE-mAb and was more pronounced with multibiotinylated LLE-mAb compared with monobiotinylated LLE-mAb.

## 4. Discussion

Strategies to improve CAR T cell therapy address both safety and efficacy. It must be underscored that to date, the main challenges in CAR T cell therapy, especially in B-lineage cancers, are not around safety [16] but their limited efficacy over a broad range of target antigens [7] and a broad range of tumor entities [13]. While early-onset toxicities can be managed satisfactorily, limited efficacy remains the unmet need [17]. Increasing the efficacy primarily corresponds to solving the antigen question. Subsequently, CAR T cell characteristics that define the antitumoral performance, such as fitness, exhaustion, and persistence [7,17,18], must be addressed in conjunction to solving tumor entity-specific environmental challenges with regards to the immunosuppressive tumor microenvironment [19,20]. Most tumors will require simultaneous targeting to successfully treat antigen-heterogeneous cancers and prevent pre-existing [21] and evolutionarily driven antigen immune escape [22]. Otherwise, the risk of treatment failure and cancer recurrence is significant due to antigen immune escape mechanisms [23].

In our current in vitro work, we demonstrated that AdCAR T cells can be used for antigen-specific targeting via adapter molecules in the presence of the corresponding antigen-positive target cells in the context of non-Hodgkin’s lymphoma cell lines. Obviously, the translational vision is to adapt CAR-mediated targeting to the antigen expression profile of the individual cancer patient by selecting adapter molecules targeted to overexpressed antigens in individual cancers, such as CD19, CD20, CD22, CD37, and CD79B, which are currently used in NHL antibody-based therapies [24,25,26]. Polyimmunotargeting should contribute to increasing the efficacy of CAR T cell therapy in non-Hodgkin’s lymphomas by preventing antigen immune escape, as has been studied with dual-targeted CAR T cells for CD19-CD22 in large B cell lymphoma patients [8] and with trispecific CD19-CD20-CD22-targeted CAR T cells in preclinical models [7]. Moreover, in our previous study, we tested combinatorial immunotargeting utilizing the AdCAR technology in the stably transduced cancerous NK cell line NK-92 with similar functional properties to AdCAR T cells [12]. The advantage of using individual T cell products compared to irradiated cancerous NK cell lines for patient use or primary NK cells are the initial exponential proliferation kinetics of CAR T cells. The fast kinetics can lead to immediate tumor control and the ability to permanently engraft in the patients and become part of the patients’ immune system, which may support long-term tumor control. CAR-engineered NK cell lines have been used in clinical trials (NCT00900809, NCT00990717, NCT02944162, and NCT02742727); however, irradiation may limit their anticancer function [27]. NK and T cell CAR effector cells can be regarded as complementary and both provide unique effector functions [28].

Alternative technologies, such as molecular on and off switches [29], or mRNA-based gene delivery for transient targeting will also allow transient polyimmunotargeting [30].

Unsurprisingly, immunophenotyping of the cell lines revealed differential antigen immune profiles for JeKo-1, Raji, and Daudi. In patients, antigen immunoprofiles must be acknowledged and addressed in targeted immunotherapies in the same way. CD19-negative relapses account for around half the number of relapses after CD19CAR T cell therapy, causing treatment failure [21]. By targeting several antigens simultaneously, the chance of a profound response is increased; however, the potency of targeting also impacts the long-term outcomes. Thus, dual- and triple-targeted CAR T cell products have been developed [7]. Adapter CAR T cell technologies allow the targeting of any surface-expressed antigen in a patient-individualized manner. Dual CD19-CD20 targeting via AdCAR T cells outperformed CD19CAR T cells using a JeKo-1^CD19KO^ variant, demonstrating the feasibility of combinatorial targeting and the superiority over single-targeted CD19CAR T cells. Combinatorial CAR targeting strategies may improve the outcome of patients by preventing the development of antigen immune escape variants [31]. 

The safety profile of AdCAR T cells is high due to its functional inactivity and its self-limiting activity, which is dependent on the pharmacokinetics and dynamics of the adapter molecules. As an example, rituximab, as a full-size antibody, is cleared from the body with a half-life of around 3 weeks [32], whereas blinatumomab (54 kDa) is cleared in (mean ± SD) 1.25 ± 0.63 h t1/2 [33]. The sophisticated elimination of CAR T cells via inducible suicide genes (iCaspase) [34] or the antibody-dependent elimination of CAR T cells by targeting a targetable co-expressed marker gene such as truncated EGFR [35] are less elegant than the functional switch in AdCAR T cells.

Moreover, adapter CAR T cells will allow the targeting of antigens that cannot be targeted by conventional CAR T cells due to dramatic on-target off-tumor toxicities [36,37]. However, most target antigens in B-lineage non-Hodgkin’s lymphoma are suitable for long-term CAR targeting [24,25,26].

In general, there are various indirect CAR approaches with distinct properties, which utilize an extracellular recognition domain that connects the intracellular CAR signaling machinery via linking molecules to cancer-associated target antigens [38]. The underlying principle in most indirect CAR systems is an exclusive artificial antibody-dependent cellular cytotoxicity. The advantage of the AdCAR technology is that it can build on FDA/EMA-approved antibodies that have been widely used in the clinic [13,39].

In our in vitro study, we used a second-generation CD19CAR construct, which is the basic architecture of all four FDA-approved CD19CAR products [2]. It remains uncertain if CD19CAR T cells would benefit from a third-generation architecture [40]. During the development of AdCAR T cells, we confirmed, in previous studies, that indirect CAR T cell constructs including AdCAR T cells and anti-FITC CAR T cells benefit from third-generation costimulatory signaling [13,41]. 

AdCAR T cells show cancer-dependent activation and cytokine secretion comparable with conventional CD19 CAR T cells. Cancer-dependent engagement is an absolute prerequisite for the application in patients. Logically, the AdCAR T cells showed superior effector function to CD19 CAR T cells in CD19-negative targets, which are a driver of relapse in patients [42]. We showed that AdCAR T cells can target different antigens and increase anticancer activity by using a combination of adapter molecules. Importantly, it was demonstrated that targeting a combination of different antigens does not lead to inhibition of the AdCAR T cells if one targeted antigen is not expressed, which was partially attributed to the favorable binding kinetics of the AdCAR scFv.

The longevity of CAR T cells is dependent on the baseline immunophenotype of a CAR T cell product [43] and the alteration in the immunophenotype and exhaustion upon stimulation of the CAR, which was comparable in the CD19 CAR T cells and the AdCAR T cells. Due to the spontaneous killing capacity of the untransduced T cells and the AdCAR T cells without the addition of adapter molecules, we analyzed the co-expression of important activating receptors in T cells and found the high expression of NKG2D and DNAM-1. In line with our assumptions, we found strong expression of the NKG2D ligands but low expression of the DNAM-1 ligands CD112 and CD155. The expression of the NKG2D ligand MIC A/B sufficiently explained the high background non-specific killing capacity of activated T cells in JeKo-1 and represented a supportive anticancer function in CAR T cells [44].

Additional genetic modifications introducing PD1-CD28 switch receptors [45] or PD1 knockout [46] could increase the potency of AdCAR T cells while still retaining the safety of the CAR T cell product. As we have underscored the importance of multitargeting, there are further implications with AdCAR T cells compared with conventional CAR T cell products. In contrast to conventional CAR T cells, CAR T cell tuning appears to be much safer due to the controllable CAR engagement and may allow us to generate more possible potent CAR T cell products than with conventional CAR T cell technologies. In conclusion, we demonstrated the advantage of multitargeting with AdCAR T cells over conventional CD19CAR T cells. Additional preclinical studies involving patient-derived xenograft models to mimic the heterogeneity of lymphomas in the naturally occurring form will demonstrate its translational potential to overcome immune escape.

## Figures and Tables

**Figure 1 biomedicines-10-02420-f001:**
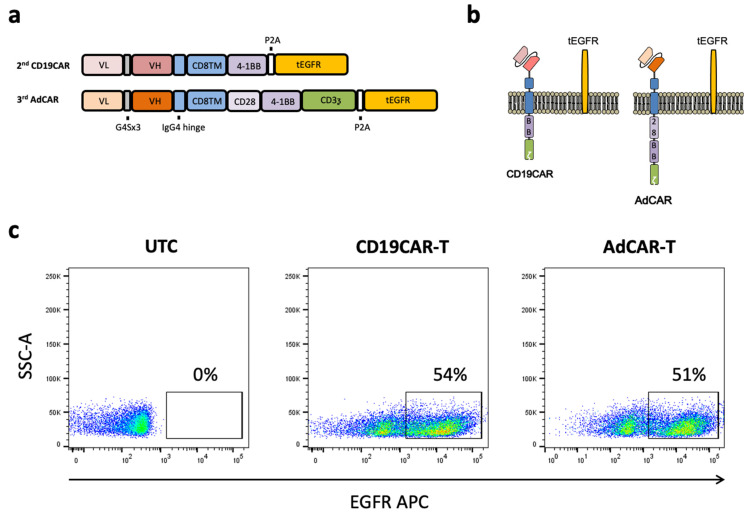
Design and expression profile of second-generation CD19CAR and third-generation AdCAR T cells. (**a**) Illustration of the transgene design of the conventional second-generation CD19CAR and the third-generation AdCAR including the marker gene. (**b**) Schematic illustration of the transgene CAR and marker gene expression in the context of the cell membrane. (**c**) The expression of CAR was determined by flow cytometry utilizing the co-expressed marker gene tEGFR. Representative flow cytometric plots for non-transduced activated T cells (**left**), CD19CAR T cells (**middle**), and AdCAR T cells (**right**) are shown.

**Figure 2 biomedicines-10-02420-f002:**
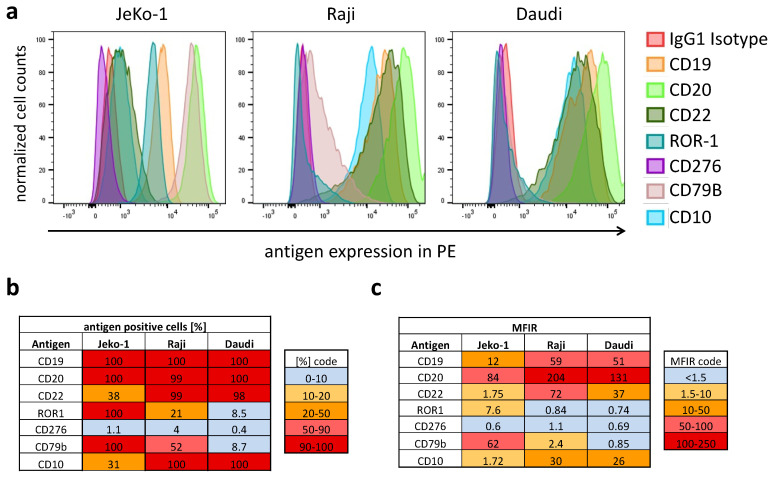
Antigen expression profile of the lymphoma cell lines JeKo-1, Raji, and Daudi. (**a**). JeKo (**left**), Raji (**middle**), and Daudi (**right**) were investigated for the surface expression of the target antigens CD19, CD20, CD22, ROR-1, CD276, CD79B, and CD10. (**b**) The fraction of antigen-positive cells (as a percentage) was calculated by Overton subtraction of the histograms. The table shows the percentage of corresponding antigen positive cells. The sample data minus the corresponding isotype control are shown in the heat map. (**c**) The relative expression level of the screened antigens is shown in the heat map as an index calculated from the median fluorescence intensity of the sample divided by the median fluorescence intensity of the isotype control, defined as the MFIR.

**Figure 3 biomedicines-10-02420-f003:**
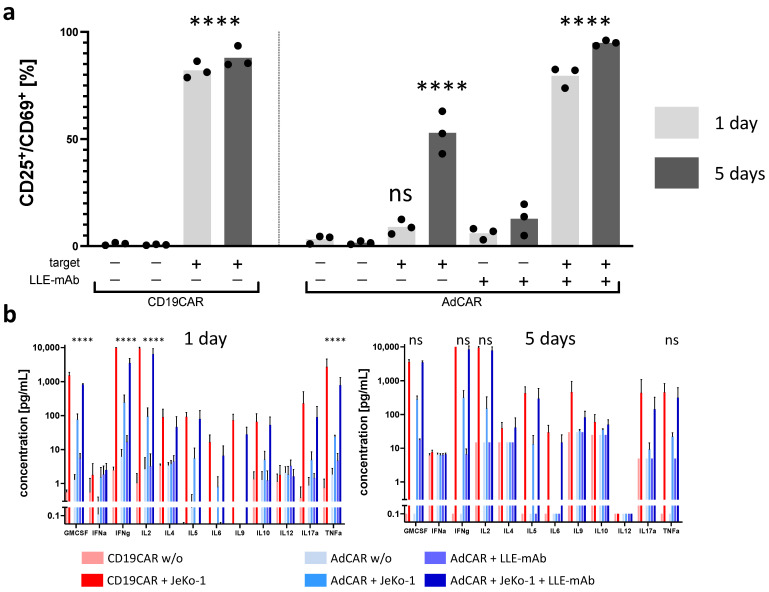
Activation and cytokine secretion profile of AdCAR T cells vs. CD19CAR T cells. (**a**) CD19CAR T cells were incubated for 24 h (light grey) and for 5 days (dark grey) with or without the CD19^+^ lymphoma cell line JeKo-1. Additionally, AdCAR T cells were incubated for 1 day (light grey) and for 5 days (dark grey) with or without the CD19^+^ lymphoma cell line JeKo-1, and with or without 10 ng/mL LLE-CD19 mAb. Specific CAR activation after incubation was determined by flow cytometry staining of the cell surface activation markers CD25 and CD69. The double-positive CD25^+^CD69^+^ cells were considered and defined to be the activated cell subset. (**b**) The quantitative levels of the indicated secreted cytokines in the supernatant were measured after 1 day (**left**) and after 5 days (**right**). Data shown in (**a**) represent the mean ± SEM of (n = 1) independent experiments from three different donors. Data shown in (**b**) represent the mean ± SEM of (n = 1) independent experiments in duplicate from three different donors. ns, not significant. ****, *p* ≤ 0.0001.

**Figure 4 biomedicines-10-02420-f004:**
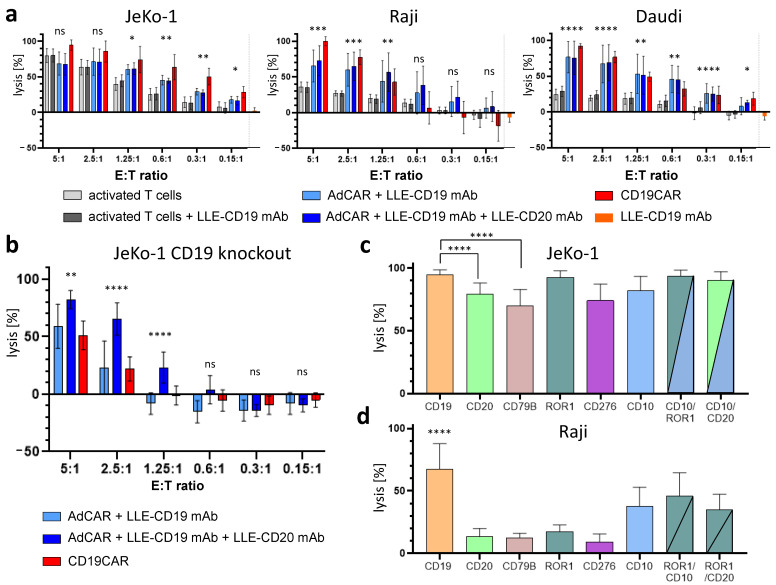
Multitargeting with AdCAR T cells prevents immune evasion in lymphoma. (**a**) Non-transduced activated T cells with or without 1 ng/mL LLE-CD19 mAb, AdCAR T cells with or without 1 ng/mL LLE-CD19 mAb +/− 1 ng/mL LLE-CD20 mAb, or conventional CD19CAR T cells were incubated with JeKo-1 (**left**), Raji (**middle**), and Daudi (**right**) at the indicated E:T ratios. Target cells only were incubated with LLE-CD19 mAb at 1 ng/mL. Target cell lysis was determined by LCA after 48 h. A statistical analysis is presented for the comparison of the AdCAR T cells + LLE-CD19 mAb versus the activated T cells. (**b**) AdCAR T cells with 1 ng/mL LLE-CD19 mAb or 1 ng/mL LLE-CD19 mAb + 1 ng/mL LLE-CD20 mAb, or conventional CD19CAR T cells were incubated for 48 h with a CD19 knockout variant of JeKo-1 (JeKo-1_CD19KO_) at the indicated E:T ratios. Target cell lysis was determined by LCA after 48 h. A statistical analysis is presented for a comparison of the combinatorial targeting (CD19 + CD20) versus the monotargeting (CD19). (**c**,**d**) To demonstrate the flexibility of the AdCAR system, AdCAR T cells were incubated with either JeKo-1 (**c**) or Raji (**d**) at an E:T ratio of 1.25:1 with 1 ng/mL of the indicated LLE-mAbs. Alternative target antigens can be utilized by AdCAR T cells. Combinatorial targeting was feasible and increased the engagement of the AdCAR T cells with the cancer cells. Target cell lysis was determined by LCA after 48 h. The statistical analysis demonstrates the superiority of CD19 as a target antigen compared with the alternative antigens CD20 and CD79B in JeKo-1. Targeting CD19 mediated the strongest cytolytic effect in Raji. Data shown in (**a**,**b**) represent the mean ± SEM of (n = 3) independent experiments in triplicate from three different donors. Data shown in (**c**,**d**) represent the mean ± SEM of (n = 3) independent experiments from three different donors. Statistical analysis was performed using one-way ANOVA and Tukey’s post-hoc test. ns, not significant. *, *p* ≤ 0.05. **, *p* ≤ 0.01. ***, *p* ≤ 0.001. ****, *p* ≤ 0.0001.

**Figure 5 biomedicines-10-02420-f005:**
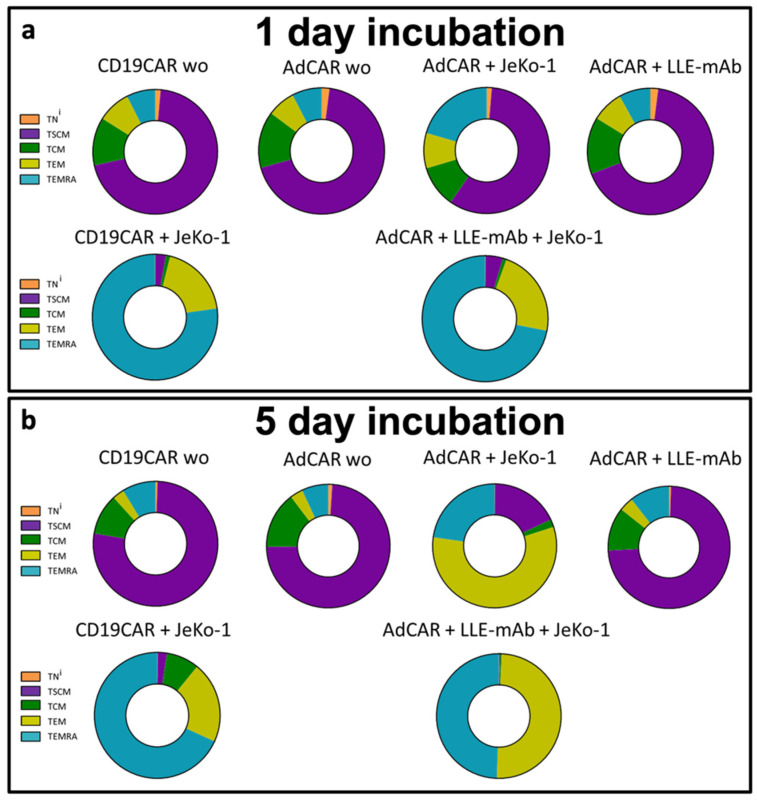
Immunophenotype of AdCAR T cells vs. CD19CAR T cells. CD19CAR T cells were incubated for (**a**) 1 day and (**b**) for 5 days with or without the CD19^+^ lymphoma cell line JeKo-1. Additionally, AdCAR T cells were incubated for (**a**) 24 h and (**b**) for 5 days with or without the CD19^+^ lymphoma cell line JeKo-1, and with or without 10 ng/mL LLE-CD19 mAb. Immunophenotypes after incubation were determined by flow cytometry by staining the cell surface markers CD45RA, CD45RO, CD62L, and CD95. Naive T (T_N_^i^) cells were defined as (CD45RA^+^/CD45RO^−^/CD62L^+^/CD95^−^), stem cell-like memory T cells (T_SCM_) as (CD45RA^+^/CD45RO^−^/CD62L^+^/CD95^+^), central memory T cells (T_CM_) as (CD45RA^−^/CD45RO^+^/CD62L^+^/CD95^+^), effector memory T cells (T_EM_) as (CD45RA^−^/CD45RO^+^/CD62L^−^/CD95^+^), and effector memory T cells that re-expressed CD45RA (T_EMRA_) as (CD45RA^+^/CD45RO^−^/CD62L^−^/CD95^+^). Data shown in (**a**,**b**) represent the mean ± SEM of (n = 3) independent experiments from three different donors.

**Figure 6 biomedicines-10-02420-f006:**
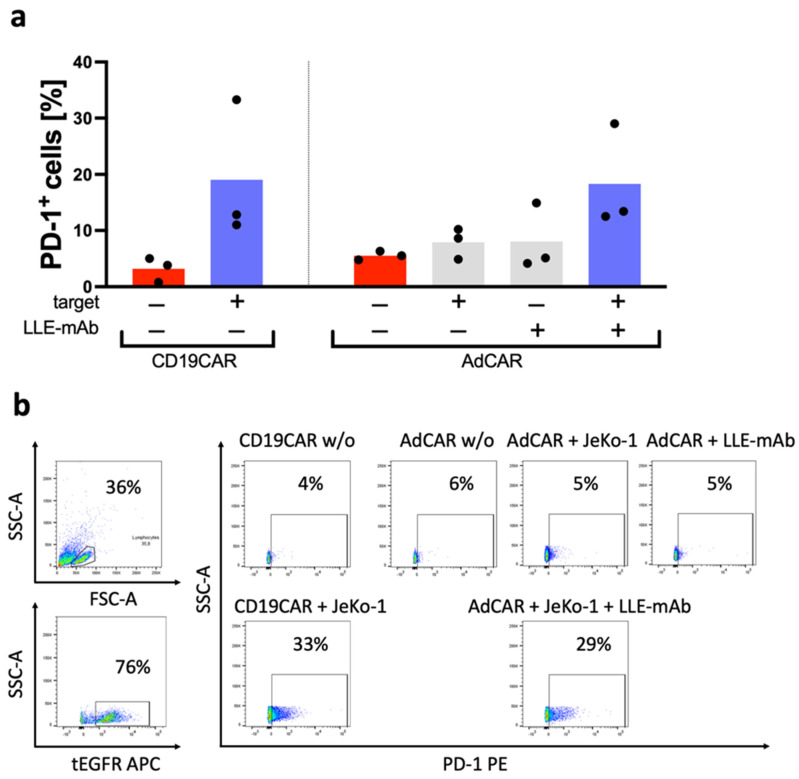
Exhaustion of AdCAR T cells vs. CD19CAR T cells. (**a**) CD19CAR T cells were incubated for 24 h with or without the CD19^+^ lymphoma cell line JeKo-1. Additionally, AdCAR T cells were incubated for 24 h with or without the CD19^+^ lymphoma cell line JeKo-1 and with or without 10 ng/mL LLE-CD19 mAb. Exhaustion after incubation was determined by flow cytometry using a PE-labeled PD-1 (CD279) mAb. (**b**) Representative schematic gating strategy of the indicated conditions of CD19CAR and AdCAR expressing cells with and without the target cell line JeKo-1, as well as with or without the LLE-mAb for the AdCAR T cells. PD-1 expression was analyzed and is illustrated for the CAR-positive cells (**left**). Data shown in (**a**) represent the mean ± SEM of (n = 3) independent experiments from three different donors.

**Figure 7 biomedicines-10-02420-f007:**
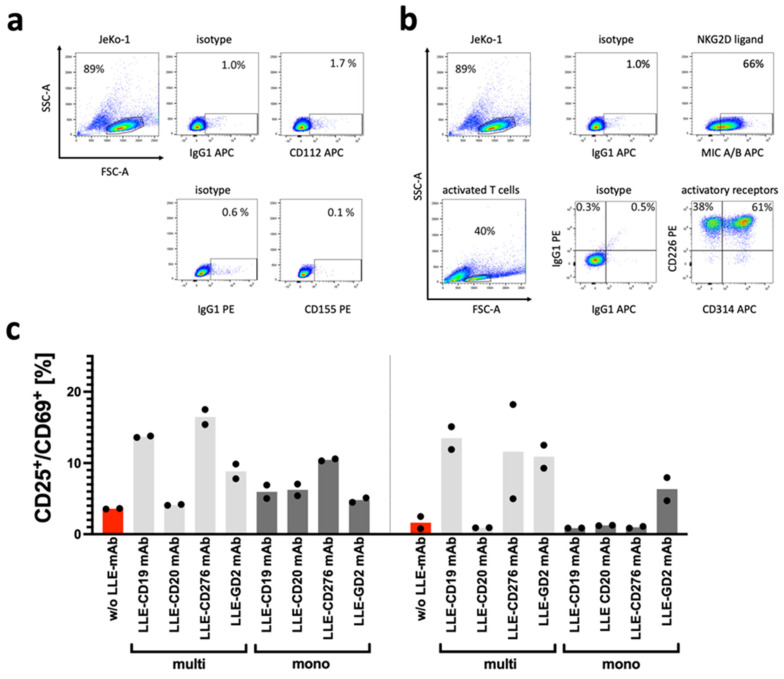
Non-specific activation of AdCAR T cells. (**a**) The surface expression of the DNAM-1 ligands CD112 and CD155 on JeKo-1 was determined by flow cytometry. (**b**) The surface expression of the NKG2D ligands MIC A/B on JeKo-1 and the surface expression of the activating receptors DNAM-1 (CD226) and NKG2D (CD314) on activated T cells was determined by flow cytometry. (**c**) The number of available binding sites for the AdCAR T cells per adapter molecule has an impact on the activation of the AdCAR T cells. To demonstrate that the non-specific activation of AdCAR T cells was enhanced in multi-biotinylated adapter molecules, mono- versus multi-biotinylated adapter molecules were investigated to induce CD25^+^CD69^+^ expression as an indicator of CAR-mediated activation of AdCAR T cells. AdCAR T cells were co-incubated without LLE-mAb (red) and with different LLE-mAbs at 10 ng/mL. The adapter molecules in full-size antibody format were targeted to CD19, CD20, CD276, and GD2. The specific activation status of the CAR T cells was determined by flow cytometry. Data shown in (**b**) represent the mean ± SEM of (n = 2) independent experiments from two different donors.

## Data Availability

All data associated with this study are presented in the paper.

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
