# Peer review of "Adapter CAR T Cell Therapy for the Treatment of B-Lineage Lymphomas"

_biomedicines, 2022, doi:10.3390/biomedicines10102420_

Round 1
Reviewer 1 Report
This research group demonstrated that AdCAR T platform can target different antigens and increase anticancer activity by using a combination of adapter molecules. The prospect of the flexibly of the proposed platform to shift the CAR specificity by exchanging the adapter molecule specificity, obviously broadens the range of individualised applications for such established immunotherapies. Further clinical evaluation of the AdCAR T platform in animals and then hopefully in lymphoma patients, should be performed to assess the therapeutic responses to overcome treatment failures owing to antigen immune escape i.e. a disadvantage of monotargeted conventional CAR T cell therapies. Polyimmunotargeting will also be assisted in the future by novel mRNA manufacturing technologies via the application of CRISPR technologies, also beyond the surrogate cell line Knock-out and this could be mentioned in the discussion see ref https://doi.org/10.3390/pharmaceutics13091371. In conclusion, this paper contributes significant novel knowledge to already proven extremely important field, this of chimeric antigen receptor adaptive therapies, thus offering to the scientific society invaluable insights to the future of personalised applications for these patients in need. Congratulations to the authors.
Author Response
Dear Reviewer,
thank you for your time to review our manuscript.
I added a sentence in the manuscript to appreciate that mRNA technologies are emerging and can also be used in the context of transient polyimmunotargeting.
Alternative technologies, such as molecular ON- and OFF-switches29, or mRNA based gene delivery for transient targeting will also allow transient polyimmunotargeting30.
Reviewer 2 Report
The authors developed an approach that allows targeting cytotoxic T cells to a wide range of tumor cell antigens, including several at the same time. The manuscript presents a study proving the effectiveness of the approach proposed by the authors. The article may certainly be of interest to readers of biomedicines. There are no significant comments on the manuscript. The work is a continuation of the studies performed by the authors on the AdCAR NK-92 cell model. The manuscript would certainly benefit if the authors discussed in more detail how the approach proposed in this paper is superior to the previously proposed model.
Author Response
Dear Reviewer,
thank you for your time to review our manuscript.
I added a paragraph on NK CAR T cells with reference to our previous work using the AdCAR technology in stably transduced NK-92.
Moreover, in our previous study we have tested combinatorial immunotargeting utilizing the AdCAR technology in the stably transduced cancerous NK cell line NK-92 with similar functional properties as in AdCAR T cells12. The advantage of using individual T cell products compared to a irradiated cancerous NK cell lines for patient use or primary NK cells are the initial exponential proliferation kinetics of CAR T cells. The fast kinetics can lead to immediate tumour control and the ability to permanently engraft in the patients and become part of the patients’ immune system may support long-term tumour control. CAR-engineered NK cell lines are used in clinical trials [NCT00900809, NCT00990717, NCT02944162, and NCT02742727], however irradiation may limit their anticancer function27. NK- and T-cell CAR effector cells shall be regarded complementarily and both provide unique effector fucntions28.